# Co-option of neck muscles supported the vertebrate water-to-land transition

Eglantine Heude [1,2] ✉, Hugo Dutel [3,4,5], Frida Sanchez-Garrido [2], Karin D. Prummel [6,7,15], Robert Lalonde [7,16], France Lam [8], Christian Mosimann [6,7], Anthony Herrel [9,10,11,12] & Shahragim Tajbakhsh [13,14]

A major event in vertebrate evolution was the separation of the skull from the pectoral girdle and the acquisition of a functional neck, transitions that required profound developmental rearrangements of the musculoskeletal system. The neck is a hallmark of the tetrapod body plan and allows for complex head movements on land. While head and trunk muscles arise from distinct embryonic mesoderm populations, the origins of neck muscles remain elusive. Here, we combine comparative embryology and anatomy to reconstruct the mesodermal contribution to neck evolution. We demonstrate that head/trunk-connecting muscle groups have conserved mesodermal origins in fishes and tetrapods and that the neck evolved from muscle groups present in fishes. We propose that expansions of mesodermal populations into head and trunk domains during embryonic development underpinned the emergence and adaptation of the tetrapod neck. Our results provide evidence for the exaptation of archetypal muscle groups in ancestral fishes, which were co-opted to acquire novel functions adapted to a terrestrial lifestyle.

The conquest of the terrestrial realm during the Carboniferous (360 MYA) represented a critical step in the evolution of vertebrates[1]. This major transition was accompanied by the emergence of key morphological novelties, including the transformation of fins into complex limbs and the emergence of a mobile neck, traits that are specific to tetrapods[2,3]. The transition from an immobile connection between the head and pectoral girdle in fishes to a free-moving head in tetrapods involved profound musculoskeletal rearrangements at the head/trunk transition during vertebrate evolution[3].

A functional neck allows orientation of the head independently from the trunk towards sensory cues and it plays a critical role in efficient feeding, respiration, and vocalisation by housing the hyoid, pharynx, and larynx components[4]. Extant tetrapods possess a complex network of four main evolutionarily conserved neck muscle groups: (1) pharyngeal and laryngeal muscles located deep in the neck; (2) hypobranchial muscles that connect to the hyoid bone ventrally; (3) dorsal epaxial and hypaxial muscles that surround the cervical vertebrae; and (4) superficial cucullaris-derived muscles that overlay the

[1]Institut de Génomique Fonctionnelle de Lyon, École Normale Supérieure de Lyon, CNRS UMR5242 Université Claude Bernard Lyon-1, Lyon, France. [2]PHYMA, Département Adaptations du Vivant, Muséum national d'Histoire naturelle, CNRS UMR 7221, Paris, France. [3]Bristol Palaeobiology Research Group, School of Earth Sciences, University of Bristol, Bristol, UK. [4]Université de Bordeaux, CNRS, MCC, PACEA, UMR 5199, Pessac, France. [5]Craniofacial Growth and Form, Hôpital Necker - Enfants Malades, Paris, France. [6]Department of Molecular Life Sciences, University of Zurich, Zurich, Switzerland. [7]Department of Pediatrics, Section of Developmental Biology, University of Colorado School of Medicine, Anschutz Medical Campus, Aurora, CO, USA. [8]Core Facilities - Institut de Biologie Paris Seine (IBPS), Sorbonne Universités, Paris, France. [9]MECADEV, Département Adaptations du Vivant, Muséum national d'Histoire naturelle, CNRS UMR 7179, Paris, France. [10]Department of Biology, Evolutionary Morphology of Vertebrates, Ghent University, Ghent, Belgium. [11]Department of Biology, University of Antwerp, Wilrijk, Belgium. [12]Naturhistorisches Museum Bern, Bern, Switzerland. [13]Department of Developmental & Stem Cell Biology, Stem Cells & Development Unit, Institut Pasteur, Université Paris Cité, Paris, France. [14]CNRS UMR3738, Institut Pasteur, Paris, France. [15]Present address: Molecular Systems Biology Unit, European Molecular Biology Laboratory (EMBL), Heidelberg, Germany. [16]Present address: Yale University, New Haven, USA. ✉e-mail: eglantine.heude@cnrs.fr

neck musculature dorso-laterally[4]. Pharyngeal and laryngeal muscles are mainly involved in food intake, respiration, and vocalisation together with hypobranchial muscles, whereas dorsal epaxial and hypaxial musculature as well as cucullaris-derived muscles predominantly coordinate head mobility and locomotion[5,6].

Deciphering the developmental origins of neck muscles is central to understanding the mechanisms that underpinned the emergence and adaptation of a functional neck in tetrapods. However, the embryonic origins of the different components composing the neck musculoskeletal system remained unclear, being at the interface of head and trunk domains[3,4,7–15].

Head and trunk muscles arise from different mesodermal progenitors in vertebrates, and their development is governed by distinct myogenic programs[16,17]. At the cranial level, a mesodermal progenitor field termed the cardiopharyngeal mesoderm (CPM) gives rise to head muscles and the second heart field[17]. CPM specification is initiated by the activation of a set of genes including *Mesp1*, *Islet1*, and *Tbx1*[4,16,17]. Loss of *Tbx1* in zebrafish and mouse embryos causes hypoplasia of branchial arch muscles and heart defects consistent with a role of *Tbx1* in CPM specification[4,14,18–20]. Posterior to the CPM, *Pax3* is expressed in the somitic mesoderm that gives rise to trunk and fin/limb muscles[21–23]. *Pax3* mutant mice have epaxial and hypaxial (including limb) muscle defects[21,23]. In zebrafish, *pax3* and *pax7* genes are redundantly required for myogenesis of migrating muscle progenitors that form fin and ventral hypaxial muscles[24]. After the distinct specification of cranial and trunk muscle progenitors, both lineages activate myogenic regulatory factors (MRFs) and myofiber markers such as different myosin heavy chains (MyHC) for myogenic commitment and differentiation[16].

In the neck, the somitic origin of the epaxial and hypaxial (including hypobranchial) musculature has been well documented by expression, grafting, and functional experiments[11,21,23,25]. More recently, laryngeal muscles were demonstrated to be of CPM origin by lineage tracing and functional analyses in the mouse[4,26]. In contrast, the embryological origin of cucullaris-derived muscles has remained controversial due to the location of its embryonic precursor at the interface of cardiopharyngeal, somitic, and lateral plate mesoderm populations[3,4,6–15,27–29]. The cucullaris is a generic term defining putative homologous muscles, innervated by the vagal nerve (cranial nerve CN X), that connect the skull to the pectoral girdle in fishes (also known as the protractor pectoralis[5]) and amphibians[13,29]. In amniotes, the cucullaris represents the embryonic anlage that gives rise to the trapezius and sternocleidomastoid muscles innervated by the accessory nerve (CN XI) derived from the vagus nerve (CN X)[4,6,29]. Depending on the species analysed and the experimental approach, the cucullaris derivatives were reported to originate either from CPM, somites, or trunk lateral plate mesoderm[9,12–14]. However, lineage and functional analyses in mouse indicate that the trapezius and sternocleidomastoid muscles originate from the CPM[4,13,15].

In fishes, ontogenetic morphological analyses in teleost and non-teleost actinopterygians also suggest that the putative cucullaris homologue (protractor pectoralis) is of CPM origin[5,10,30]. However, the embryonic genetic origin of the fish putative cucullaris has not been assessed, despite its importance for reconstructing the ancestral neck myogenic programs in bony vertebrates (actinopterygians + sarcopterygians) and deciphering the origin the tetrapod neck.

Here, we perform transgenic lineage tracing in zebrafish embryos to map the relative contributions of the cardiopharyngeal and somitic mesoderm to the head-trunk transition. We show that muscle groups connecting the head and trunk in teleosts and tetrapods share conserved mesodermal lineage origins in the embryo. Based on this ancestral developmental blueprint of jawed vertebrates, we then use 3D high-resolution scans for comparative anatomical analyses to infer the relative contribution of cardiopharyngeal and somitic mesoderm to neck musculature in living taxa spanning the fish-tetrapod transition including ray-finned fishes, lobe-finned fishes, salamanders, and

lizards. Our combined data demonstrate that neck muscle groups predated the emergence of tetrapods. We propose that archetypal muscle groups in ancestral fishes were co-opted for new functions that facilitated the adaptation to land in early tetrapods.

## Results
### Conserved mesodermal origins of muscle groups at the head/trunk transition

The identity of skeletal muscles is generally defined by three factors: the embryonic mesodermal origin, the innervation, and the attachments to skeletal structures. We used these criteria to determine which and how different mesodermal populations contributed to the emergence and adaptation of the neck.

We first sought to assess the homology of muscles that connect the head and trunk between zebrafish and tetrapods by studying their embryonic origins and innervation. We applied transgenic *pax3a:EGFP* and *tbx1:creERT2* reporter zebrafish as complementary markers to map the contribution of the somitic and cardiopharyngeal mesoderm respectively, to the head/trunk-connecting muscles (HTM).

Zebrafish harbour two *pax3* paralogous genes, *pax3a* and *pax3b* with differential expression within somitic mesoderm. During early development, *pax3a* is expressed in the neural tube and in the somites along the entire axis while *pax3b* is restricted to the hypaxial region of the anterior somitic mesoderm[31]. As observed for *Pax3* in the mouse embryo, zebrafish *pax3a* is expressed during muscle progenitor specification, then decreased in differentiating myofibers. The transgenic *pax3a-EGFP* zebrafish reporter exhibits perdurance of GFP in myofibers for several days, rendering it a valuable tool for temporally monitoring the somitic lineage[22]. We analyzed the spatiotemporal expression of *pax3a-EGFP* from 1 to 5 days post-fertilization (dpf) to track HTM development until the emergence of the putative cucullaris, which is the last muscle to differentiate by activating MyHC at the head/trunk transition[5] (Fig. 1a–o).

Throughout all stages analysed, we observed strong GFP labelling in the neural tube and in hypaxial muscles of migratory origin including the ventral hypaxial, the pectoral fin, and the hypobranchial sternohyoid muscles (Fig. 1, green arrowheads). In contrast, no GFP was detected in the branchial arch muscles (Fig. 1a–l), the cucullaris (Fig. 1m–o), and the coracobranchial muscles (Fig. 1p–r), the latter connecting the branchial series to the cleithrum (a part of the pectoral girdle) in fishes[32] and being absent in tetrapods.

To map the muscles emerging from the *tbx1*-expressing CPM in zebrafish, we performed genetic lineage tracing using inducible Cre driver *tbx1:creERT2*[19,33] embryos (Fig. 2a–l, Supplementary Fig. 1). *tbx1:creERT2;hsp70l:Switch* embryos were induced with 4-OH-Tamoxifen (4-OHT) at shield stage (6 h post-fertilization, hpf) to 30 hpf, and the resulting *hsp70l:Switch*-dependent GFP-lineage labelling were analysed at 5 dpf. Consistent with previous work[19], anterior lateral plate mesoderm and CPM derivatives were labelled, including cardiomyocytes (Fig. 2c, blue arrowhead) and branchial arch muscles (Fig. 2c, green arrowheads). The cucullaris myofibers showed reproducible positive *tbx1:creERT2* lineage labelling, along with the posterior branchial levator and coracobranchial muscles (Fig. 2a–i, Supplementary Fig. 1, green arrowheads). In contrast, and consistent with specific CPM-focused muscle lineage labelling by *tbx1:creERT2*, GFP expression was not detected in adjacent somite-derived hypaxial musculature of migratory origin (Fig. 2g–i, Supplementary Fig. 1a–c, note the vhp/pfm/sh muscles). Combined with the *pax3a-GFP* reporter analysis, we conclude that the cucullaris muscle in zebrafish does not originate from somitic mesoderm, but originates from CPM as reported for mice[4,15].

To further test this homology, we analysed the innervation pattern of these muscles in zebrafish larvae and axolotl larvae as a tetrapod reference (Fig. 2j–l, Supplementary Figs. 2, 3, Supplementary Movies 1–3). The developing cucullaris of 5 dpf zebrafish larvae resided

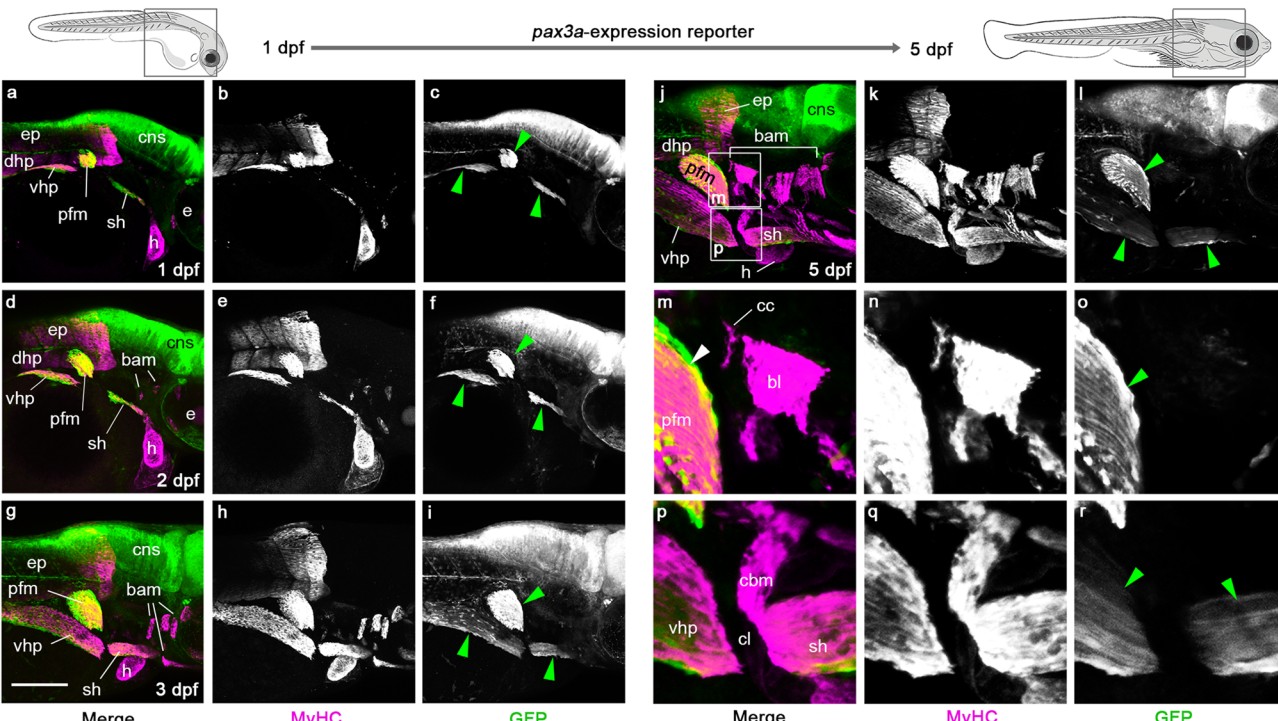

**Fig. 1 | Somitic mesoderm contribution to head/trunk-connecting muscles in the zebrafish larva.** Whole-mount immunofluorescent stainings showing the dynamics of *pax3a* reporter expression (GFP, green) in head/trunk-connecting muscles (MyHC, magenta) of *pax3a-GFP* zebrafish larvae from 1–5 days post-fertilization (dpf) at the level indicated on schemes. Anti-Myosin Heavy Chain (MyHC) immunofluorescence and *pax3a-GFP* expression are shown separately in grey levels in **b**, **e**, **h**, **k**, **n**, **q** and **c**, **f**, **i**, **l**, **o**, **r** respectively. **m**–**p** Higher magnifications of regions indicated in **j**. From 1 dpf to 5 dpf, *pax3a-GFP* expression is observed in developing muscles of migratory somitic origin including pectoral fin muscles

(pfm), ventral hypaxial muscles (vhp) and sternohyoid muscles (sh) (green arrowheads), but not in branchial arch muscles (bam) and heart (**a**–**l**). Note, no GFP expression is detected in coracobranchial muscle (cbm) nor cucullaris muscle (cc) while differentiation starts at 5 dpf (**m**–**r**) (*n* = 15 independent specimens analysed for each stage). bam branchial arch musculature, bl 5th branchial levator muscle, cbm coracobranchial muscle, cc cucullaris muscle, cl cleithrum, cns central nervous system, dhp dorsal hypaxial musculature, dpf days post-fertilization, e eye, ep epaxial musculature, h heart, pfm pectoral fin musculature, sh sternohyoid muscle, vhp ventral hypaxial musculature. Scale bar in **g**, for a-l 200 μm, for m-r 50 μm.

alongside the vagus nerve (CN X) (Fig. 2j, k, Supplementary Fig. 2a–d, Supplementary Movie 1), that later connects the muscle during neuromuscular differentiation at 7dpf, as also observed at larval stage in axolotl (Supplementary Fig. 2e–l, Supplementary Fig. 3, Supplementary Movies 2, 3). The vagus nerve also innervates the ventrally located coracobranchial muscles, while the sternohyoid muscle is innervated by the hypoglossal nerve (CN XII)[3,6,29] (Fig. 2j, l, Supplementary Fig. 2, Supplementary Movie 1). Of note, the embryonic origin of fish coracobranchial muscle, also positioned at the head/trunk transition akin to the cucullaris muscle, was previously unclear; both somitic and cardiopharyngeal mesoderm origins were suggested[6,32,34]. Our combined lineage and innervation data are consistent with a cardiopharyngeal origin of coracobranchial muscles (Fig. 1p–r, Fig. 2g–i, l, o).

Altogether, our genetic lineage tracing and anatomical observations support the presence of a cucullaris homologue in zebrafish (Figs. 1, 2, Supplementary Figs. 1–3, Supplementary Movies 1–3). Our results indicate that HTM likely share conserved mesodermal origins among bony vertebrates, with dorsal epaxial/hypaxial and hypobranchial musculature of somitic origin and a cucullaris muscle of cardiopharyngeal origin. However, the evolutionary trajectory of neck muscles remains to be reconstructed (Fig. 2p).

**Evolution of head/trunk-connecting muscles across vertebrates**
The conserved embryonic origin of HTM between teleost fishes and tetrapods allowed us to infer the relative contribution of cardiopharyngeal and somitic mesoderm to HTM in extant species spanning the fish-tetrapod transition. We therefore selected key species for a comparative anatomical study to obtain a wide overview of HTM rearrangements characterising jawed vertebrate evolution. The analysis was

made on one specimen for each species of interest after validation of soft tissue integrity on virtual micro-CT sections. We included the representative teleost zebrafish (*Danio rerio*), for which we collected data in the embryo, and the basal actinopterygian bichir (*Polypterus senegalus*) (Fig. 3a, b, Supplementary Figs. 4, 5). The bichir shows skull-pectoral girdle separation, uses aerial lung respiration, and can perform tetrapod-like terrestrial locomotion using its pectoral fins, making this extant fish particularly pertinent to study vertebrate terrestrialisation[35]. We also analysed the African coelacanth (*Latimeria chalumnae*) and African lungfish (*Protopterus dolloi*) as representatives of the two known extant lineages of lobe-finned fishes that also show skull-pectoral girdle separation (Fig. 3c, Supplementary Figs. 6, 7). In contrast to the living coelacanth that has lost the capacity of lung ventilation, the African lungfish is an obligate air-breather[36]. In addition, we investigated two salamander species showing a short neck with limited mobility: the paedomorphic axolotl (*Ambystoma mexicanum*) that does not undergo metamorphosis and is bound to an aquatic habitat, and the emperor newt (*Tylototriton shanjing*) that becomes terrestrial after metamorphosis (Fig. 3d, e, Supplementary Figs. 8, 9). Finally, we included the Green Anole lizard (*Anolis carolinensis*) as an amniote representative with a flexible long neck (Fig. 3f, Supplementary Fig. 10).

Ray-finned and lobe-finned fishes exhibited limited hypobranchial and cucullaris musculature compared to tetrapods (Fig. 3). In fishes, the hypobranchial musculature is composed of the sternohyoid and coracomandibular muscles that connect the pectoral girdle to the ceratohyal (hyoid arch) and the mandible (mandibular arch), respectively[5]. In agreement with a previous report[5], the hypobranchial coracomandibular component was absent in zebrafish and lungfish

(Fig. 3a, c). The coracobranchial and cucullaris-derived muscles were present in all fish species sampled but not in the coelacanth, as previously reported[37] (Fig. 3a–c, Supplementary Figs. 4–7). We also noted that a laryngeal musculature, posterior to the branchial series and connecting the entrance of the oesophagus, was present in the bichir, coelacanth, and lungfish (Fig. 3b, c, Supplementary Figs. 5–7).

Aquatic and terrestrial salamanders showed differences in hypobranchial, laryngeal, and cucullaris-derived musculature (Fig. 3d, e, Supplementary Figs. 8, 9). In the axolotl, the shape of these muscles was akin those of lobe-finned fishes. However, the coracobranchial muscles were absent and laryngeal muscles were associated with laryngeal cartilages located anterior to the oesophagus. The metamorphic newt showed a dorso-ventral extension of the cucullaris-derived muscle onto the scapula, the laryngeal muscles and cartilages were more developed compared to the paedomorphic axolotl, and the hypobranchial musculature extended into the oral cavity to form the tongue. Changes in the musculature between paedomorphic and metamorphic salamanders was associated with a reduction of the branchial skeleton.

The lizard presented a laryngeal musculoskeletal system associated to a robust hypobranchial, including tongue, musculature. Moreover, the cucullaris derivatives had an additional attachment site ventrally and differentiated into the trapezius and sternocleidomastoid muscles, which connect the scapula and sternum to the skull, respectively (Fig. 3f, Supplementary Fig. 10)[5]. In mammals, the trapezius differentiates into anterior acromiotrapezius and posterior spinotrapezius muscle processes, the latter expanding over the entire thoracic region[4].

## Discussion

Both developmental and palaeontological evidences have narrowed the morphological gaps between fish and tetrapods[1–3,10,25,35,38]. The tetrapod body plan was built by consecutive evolutionary tinkering of pre-existing traits[1,2,38]. Our comparative analyses indicate that the neck muscle groups found in tetrapods were present in some form in the fish ancestor.

Supported by our embryonic lineage data, our findings are consistent with a model where four major neck muscle groups have been elaborated and co-opted towards acquiring new functions in the neck after the water-to-land transition. These include the somitic dorsal epaxial/hypaxial and hypobranchial muscles, and the cardiopharyngeal-derived cucullaris and laryngeal musculature, that are also found among the fish species we analysed (Fig. 4). Based on the shared developmental blueprint of HTM across bony vertebrates, our data indicate that the musculoskeletal transformations associated with the acquisition of a neck during the fish-tetrapod transition entailed two developmental changes: 1) the anterior expansion of the somitic mesoderm-derived hypobranchial musculature within the oral cavity, where it forms the tongue, and 2) the posterior expansion of the CPM-derived myogenic progenitors to form the trapezius and sternocleidomastoid muscles in amniotes (Fig. 4).

Cucullaris-derived homologous muscles have been described in various jawed vertebrates, including cartilaginous and bony fishes, and also in placoderm fossils[3,39–41]. The embryonic origin of the cucullaris was ambiguous as cardiopharyngeal, somitic, or trunk lateral plate mesoderm origins were reported[9,12–14]. Our genetic analyses in zebra-

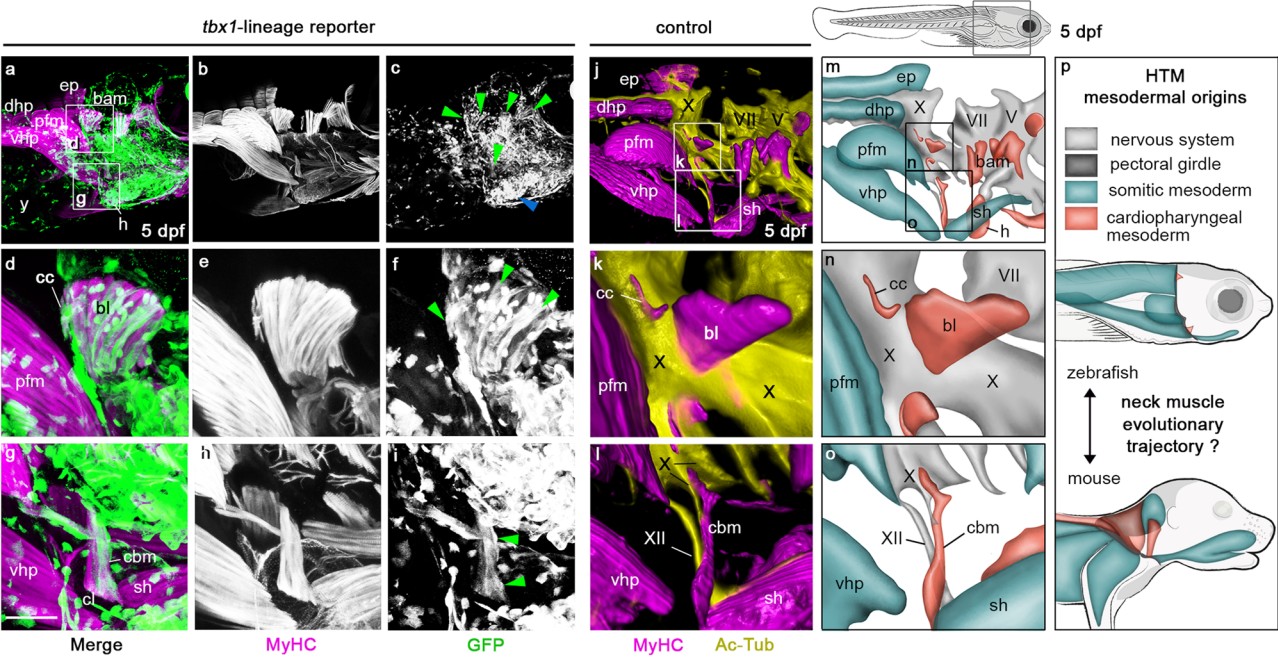

**Fig. 2 | Cardiopharyngeal mesoderm contribution to head/trunk-connecting muscles in zebrafish larva.** Whole-mount immunofluorescent staining showing *tbx1*-lineage reporter expression (GFP, green) in head/trunk-connecting muscles (MyHC, magenta) **a**–**i** and 3D rendering of muscular and nervous system (Ac-Tub, yellow) (**j**–**l**) of zebrafish larvae 5 days post-fertilization (dpf) at level indicated on scheme in **m**. Myosin Heavy Chain (MyHC) and GFP expression are shown separately in grey levels in **b**, **e**, **h** and **c**, **f**, **i**, respectively. **d**–**i**, **k**, **l** show higher magnifications of regions indicated in **a**, **j**. At 5 dpf, numerous GFP-positive myofibers are observed in branchial arch muscles (bam, green arrowheads) and in heart (h, blue arrowhead) (**a**–**c**). *tbx1*-derived myofibers are also detected in coracobranchial (cbm) and cucullaris (cc) muscles (green arrowheads) (**d**–**i**) (*n* = 10 independent specimens analysed). 3D rendering of neuromuscular system in zebrafish control larvae at 5 dpf shows that both coracobranchial (cbm) and cucullaris (cc) muscles

are innervated by branches of vagus nerve X, while sternohyoid is innervated by hypoglossal nerve XII (**j**–**l**) (*n* = 10 independent specimens analysed). **m**–**p** Schemes summarizing the data presented in Figs. 1, 2 and in ref. 4 showing somitic and cardiopharyngeal mesoderm origins and innervations of head/trunk-connecting musculature (HTM) in zebrafish larva (**m**–**o**) completed with data in mouse foetus (**p**). Colour code for **m**–**p** is indicated in **p**. bam branchial arch musculature; bl, 5th branchial levator muscle; cbm coracobranchial muscle, cc cucullaris muscle, cl cleithrum, dhp dorsal hypaxial musculature, dpf days post-fertilization, e eye, ep epaxial musculature, h heart, pfm pectoral fin musculature, HTM head/trunk-connecting muscles, sh sternohyoid muscle, vhp ventral hypaxial muscles, X vagus cranial nerve, XII hypoglossal cranial nerve, y yolk. Scale bar in **g**, for **a**–**c**, **j** 200 μm, for **d**–**l**, **k** 50 μm, for **l** 65 μm.

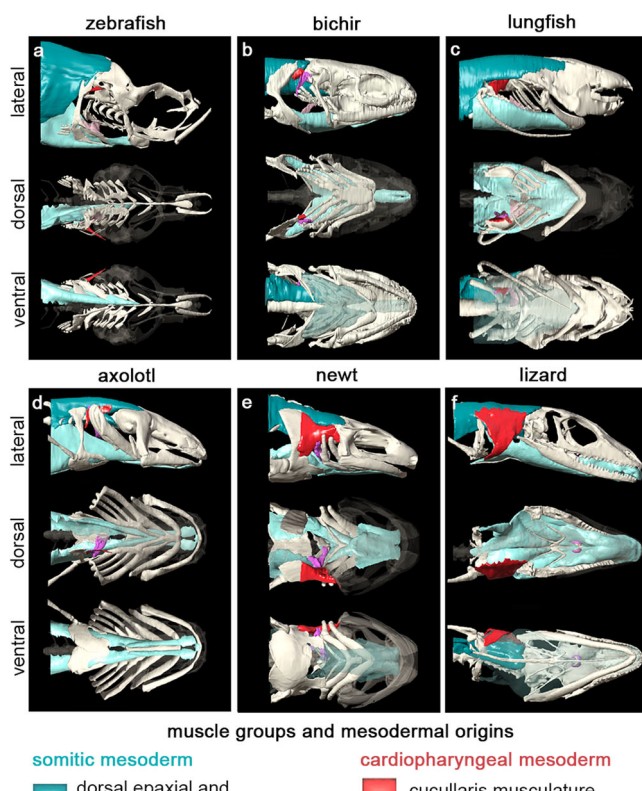

**Fig. 3 | Musculoskeletal system at the head/trunk transition in key bony vertebrates.** Lateral, ventral, and dorsal views of the 3D reconstructions of the anterior skeleton and muscle groups connecting the head to the trunk (HTM) in juvenile zebrafish *(D. rerio)* (**a**), bichir *(P. senegalus)* (**b**), lungfish *(P. dolloi)* (**c**), axolotl *(A. mexicanum)* (**d**), emperor newt *(T. shanjing)* (**e**) and green anole *(A. carolinensis)* (**f**). The dorsal epaxial/hypaxial muscles and ventral hypaxial (including hypobranchial) muscles of somitic origin are shown in dark and light blue, respectively. The cucullaris, coracobranchial, and laryngeal musculature derived from the cardiopharyngeal mesoderm are shown in red, pink, and violet respectively.

fish and mouse[4] demonstrate the conserved CPM origin of cucullaris derivatives in bony vertebrates. The data collectively indicate that the cardiopharyngeal-derived cucullaris musculature, with initial function in pectoral stabilisation in fishes, underwent successive dorso-ventral and posterior extensions into the trunk domain, assuring locomotion and head mobility in amniotes.

Notably, our data show that the bichir, coelacanth, and lungfish possess a laryngeal musculature. All of these species have lungs, and albeit only vestigial in the living coelacanth, a comparative analysis indicated that the ancestral bony fish possessed a pulmonary system[36]. Our genetic lineage data in the mouse embryo demonstrated that laryngeal muscles are CPM derivatives[4,26], but little is known about the origin of the laryngeal musculature in fish. Here, we observed that the morphology of the bichir and lungfish laryngeal muscles resemble that of the axolotl, in tight connection with the cucullaris musculature and vagus nerve, supporting homology and a CPM origin of these muscles. Although further investigation is required in both of these fish species, a CPM origin of laryngeal musculature was reported in the early-diverging Australian lungfish *Neoceratodus*[42].

Our comparative anatomical observations lead us to propose that the last common ancestor of actinopterygians and sarcopterygians possessed lungs associated with a laryngeal musculature of cardiopharyngeal origin (Fig. 4). The primitive function of laryngeal muscles has been hypothesized to assist in lung ventilation, but this function

has been lost in teleost and coelacanth lineages[36,43]. The data suggest that CPM-derived laryngeal muscles have retained their ancestral role in lung ventilation during tetrapod evolution, while being co-opted for new functions in terrestrial acoustic communication in association with the acquisition of laryngeal cartilages.

In fishes, the hypobranchial and anterior epaxial/hypaxial musculature enable suction feeding in the aquatic habitat by expanding the orobranchial cavity. Our data reveal drastic differences in hypobranchial musculature between aquatic and terrestrial salamanders. Previous work indicated that metamorphosis in salamanders is associated with modifications of the hyobranchial apparatus and changes in feeding behaviour, from a suction- to a tongue-based prey transport[44]. The cumulative data suggest that the emergence of the tongue in early tetrapods, in analogy with salamanders, contributed to the acquisition of new feeding mechanisms, thereby improving food capture, intraoral transport, and swallowing processes following the transition to land[44,45].

Exaptation refers to a mode of evolution where an existing trait is recruited or co-opted for a novel function. Broadly accepted models build on the idea that successive exaptation and adaptation has played a critical role in the evolution of form and function in vertebrates[46,47]. Our data reveal that the terrestrialisation of vertebrates went hand in hand with the exaptation of head/trunk-connecting muscles of fish ancestors to form the neck muscular system in tetrapods. Both the posterior expansion of the CPM into the trunk domain and the opposing anterior expansion of the somitic mesoderm in the oral cavity contributed to neck evolution as the skull separated further from the trunk. The resulting remodelling of skeletal, muscle, and nerve architecture greatly augmented head mobility, but also modified respiration, food intake, and vocalisation processes adapted to a terrestrial lifestyle.

## Methods
### Animals
**Transgenic zebrafish.** Zebrafish were maintained and staged as described[48] in agreement with recommendations of the national authorities of Switzerland (Animal Protection Ordinance) and following the regulation of the 2010/63 UE European directives. Protocols were approved by the cantonal veterinary office of the Canton Zurich (Kantonales Veterinäramt, permit no. 150) and by the veterinary office of the IACUC of the University of Colorado School of Medicine (protocol no. 00979), Aurora, Colorado. All zebrafish embryos were raised in temperature-controlled incubators without light cycle at 28 °C as described[48]. Established transgenic lines used in this study include *tbx1:creERT2$^{zh703}$* (Tg (−3.2tbx1:creERT2,cryaa: Venus)), *hsp70l:Switch$^{zh701}$* (Tg (−1.5hsp70l:loxP-STOP–loxP-EGFP,cryaa:Venus))[19,33], and *pax3a-GFP* (Tg(pax3a:eGFP))[22]. *tbx1:creERT2* lineage tracing experiments were performed by crossing female *hsp70l:Switch* carriers with male CreERT2 driver transgenics[33]. For CreERT2 recombinase activation, embryos were treated at shield stage (6 hours post-fertilization (hpf)) in preheated 4-OHT-containing E3 medium (Cat#H7904: Sigma) as per our established protocols[33]. 4-OHT was washed off at 30 hpf and replaced by E3 medium containing 0.003% 1-phenyl-2-thiourea (PTU, Cat#P7629; Sigma) to prevent the formation of pigmentation. Heat-shock to trigger *hsp70l:Switch* reporter expression was performed at 5 days post-fertilization (dpf) for 60 min at 37 °C in glass tubes in a water bath and larvae were raised 4-5 hours after heatshock. *pax3a:EGFP* expression analyses were performed by crossing wild-type males and *pax3a:EGFP* females and embryos were raised in temperature-controlled incubators until the desired stage. Embryos were anaesthetised in tricaine 1X (MS222, Sigma-Aldrich, A5040), equivalent to a final concentration of 160 μg/ml. EGFP-positive embryos and negative controls were fixed in 4% paraformaldehyde (PFA) at 4 °C overnight and subsequently washed and stored in PBS. Embryos from 3 independently crossed pairs were used.

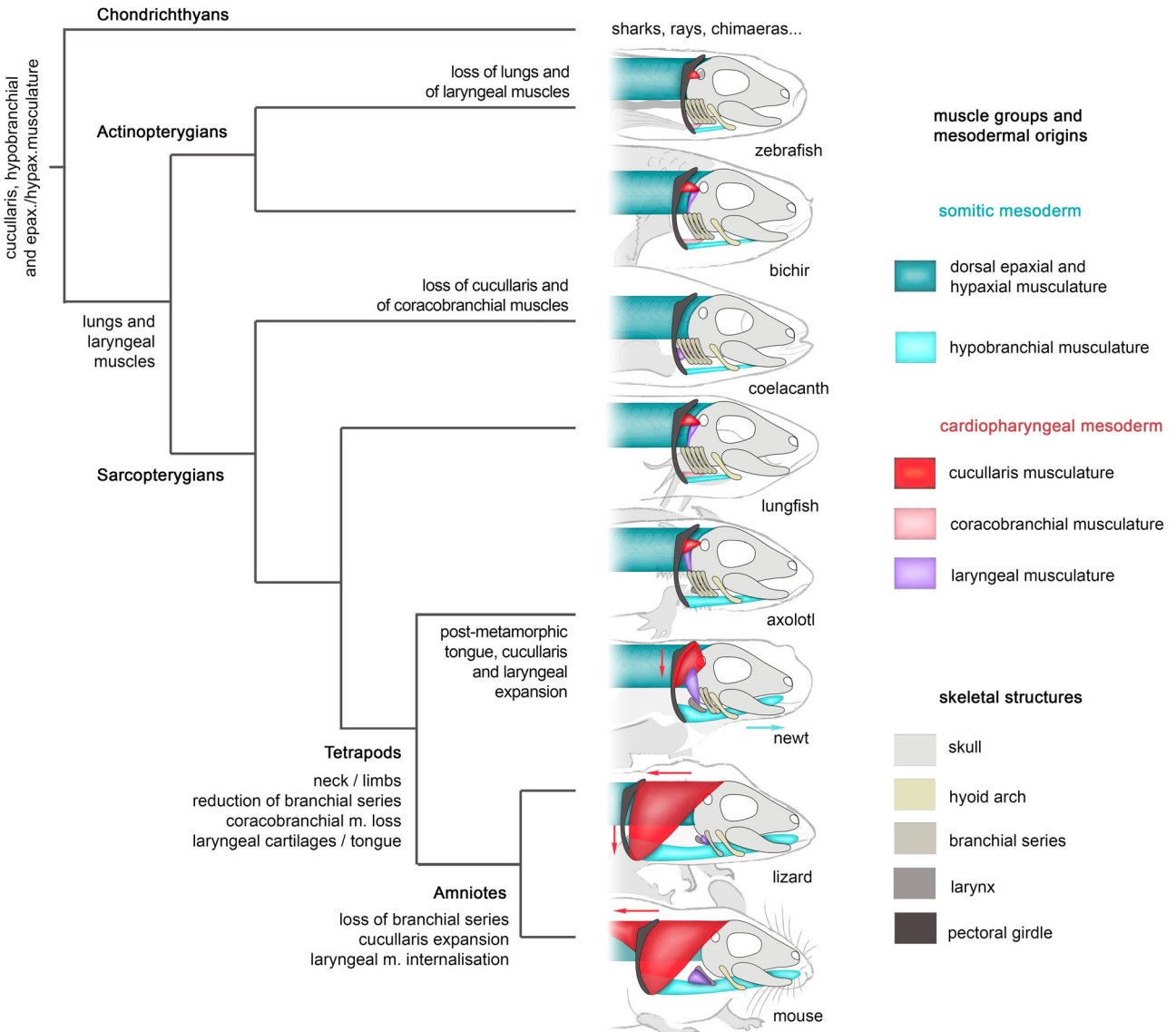

**Fig. 4 | Contributions of the somitic and cardiopharyngeal mesoderm to the emergence and adaptation of neck muscles in jawed vertebrates.** Simplified cladogram of jawed vertebrates showing the arrangement of head/trunk-connecting muscles (HTM) of cardiopharyngeal or somitic origins in the key extant vertebrate species analysed. Major evolutionary morphological traits are indicated.

**Axolotl.** Axolotls were maintained and staged as described[49] in accordance with the 2010/63 UE European directives. Axolotl larvae were generated by crossing male and female albino specimens. Larvae were maintained in water at 18 °C and fed daily. Larvae at the desired developmental stage were anesthetised in MS222, fixed in 4% PFA overnight at 4 °C and subsequently washed in PBS, dehydrated, and stored in absolute methanol.

The axolotl juvenile used for micro-computed tomography (micro-CT) scanning was euthanised four months after hatching with an overdose of MS222 in a water tank for 30 min. The specimen was then directly fixed in 4% PFA for 2 days at room temperature to preserve tissue integrity, washed in PBS and stored in 70% ethanol before contrast agent treatment.

**Other species.** The other juvenile specimens for tomography scanning were obtained from public natural history collections, no specimens were collected in the field for this project. Details on the juvenile coelacanth specimen were reported[50]. The juvenile coelacanth (ZSM 28409 (CCC 162.21)) was housed in the Bavarian State Collection for Zoology, Germany. The specimen was fixed in formalin and preserved in this fixative until 2010 before being transferred to a solution of ethanol (75%)[50]. The bichir and lungfish were obtained from the local collection at the National Museum of Natural History in Paris (MNHN, FR) and a deceased newt was obtained from our MNHN colony. Both specimens were fixed into 4% formalin solution, and then preserved in 70% ethanol solution. The precise timing of collection, fixation and storage procedures were not available for bichir, coelacanth, lungfish, newt and lizard specimens. Tissue damages may have occurred prior to analyses, but the overall soft tissue quality was validated on virtual micro-CT sections before specimen selection and structure segmentation.

Sex and gender considerations were not relevant to the biological process analysed in larval and juvenile specimens included in the study.

**Immunofluorescence staining**

Samples in absolute methanol were incubated overnight at 4 °C in Dent's fixative (20% DMSO–80% methanol) for permeabilization and

rehydrated in methanol-PBS grade series. Rehydrated samples in PBS were then blocked for 2 h in 10% normal goat serum, 3% BSA, 0.5% Triton X-100 in PBS. Primary antibodies against the green fluorescent protein (GFP, dilution 1/1000, ab13970 from Abcam), Myosin Heavy Chain (MyHC, dilution 1/20, A4.1025 from DHSB) and acetylated tubulin (ac-Tub, dilution 1/500, T7451 from Sigma-Aldrich) were diluted in blocking solution and incubated 3 days at 4 °C. After several washes in PBS with 0.1% Tween 20 (PBST), goat anti-chick Alexa 488 and goat anti-mouse Alexa 555-IgG2a and Alexa 647-IgG2b secondary antibodies (dilution 1/500, Invitrogen A-11039, A-21137, A-21242) were diluted in blocking solution and incubated 2 days at 4 °C. After whole-mount immunostaining, zebrafish larvae were washed in PBST, post-fixed in PFA 4% overnight at 4 °C and store in PBST. For confocal and light-sheet acquisitions, larvae were clarified according to the ECi protocol as described[51].

### Immunofluorescence acquisitions

Confocal images were acquired using Zeiss LSM 700 and LSM 980 laser-scanning upright confocal microscopes with dry 10x NA 0.25 and water immersion 20x NA 0.75 Zeiss objectives and ZEN software (Carl Zeiss, Germany). Light-sheet images were acquired with the Alpha3 system (PhaseView, France) with a pixel size of 650 nm and a z-step size of 1 μm. The microscope was equipped with two dry 10x NA 0.25 Zeiss objectives for a simultaneous illumination on the right and left side, the detection was performed with an 10x NA 0.60 Olympus objective with a correction ring for refractive index matching. For the blue, red and far-red channels, the 405 nm, 561 nm, and 633 nm laser lines were used for the excitation respectively. To have the same light-sheet thickness all along the x-axis, the system was equipped with the "real time focus sweeping" option. For 3D rendering, acquired Z-stacks were 3D reconstructed using the Arivis software. All images were assembled in Adobe Photoshop (Adobe Systems, USA).

### Tomography scan acquisitions and analysis

The synchrotron CT scan of the zebrafish juvenile (33 dpf) treated with phosphotungstic acid (PTA) was publicly available from the Chen Laboratory repository[52]. After fixation, *Anolis*, *Ambystoma* and *Protopterus* specimens were treated with 2.5 to 5% PMA (phosphomolybdic acid) contrast agent and micro-CT scanning was performed at the XTM Facility (Palaeobiology Research Group, University of Bristol) following the parameters reported in (Supplementary Table 1). *Tylototriton*, *Polypterus* and *Latimeria* specimens were scanned using long distance phase-contrast X-ray synchrotron tomography on BM19 at the European Synchrotron Facility following the parameters reported in (Supplementary Table 2). Segmentation was performed using Mimics Innovation Suite 19 software (Materialise SV). After segmentation, STL files were imported in Blender (Blender Foundation) for rendering.

### Reporting summary

Further information on research design is available in the Nature Portfolio Reporting Summary linked to this article.

## Data availability

All data supporting the findings of this study are available within the paper and its Supplementary Information.

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

## Acknowledgements

We acknowledge Emi Murayama and Philippe Herbomel for the kind gift of *pax3a*-GFP reporter zebrafish and Bhart-Anjan S. Bhullar and Matteo Fabbri for providing the Anolis micro-CT scan. We are grateful to Phi-lippe Durand and Jean-Paul Chaumeil for maintaining the axolotl colony at the MNHN (FR). We thank the imaging platforms of Institut Pasteur and the IBPS (FR), the micro-CT scan platform of the University of Bristol (UK) and the European Synchrotron Radiation Facility (ESRF, FR). We thank Elizabeth G. Martin-Silverstone for her technical assistance while per-forming the CT acquisitions. We thank Paul Tafforeau for his expertise and assistance for performing the synchrotron acquisitions. We also thank Florent Goussard (MNHN, FR) for initial assistance for CT scan analysis with Mimics and Julie Gamart for illustrations (J.Gam'ART, FR). We are grateful to Neil Shubin, Thibaut Brunet, Michalis Averof, Abder-rahman Khila and François Leulier for their insightful comments and discussions. We would like to highlight also the work of Dumas et al. (2024) *Development* Aug. 15;151(16) on retinoic acid and neck muscle development that appeared while our manuscript was in revision stage. This work was supported by the National Museum of Natural History (MNHN) and the Agence Nationale de la Recherche (ANR-21-CE13-0025) (E.H.); H.D. was founded by NERC (Natural Environment Research Council) Standard Grant NE/P013090/1; beamtime from the ESRF to perform the PPC-SRµCT acquisitions (proposal EC-1023) (H.D.); the Swiss National Science Foundation (SNSF) professorship (PPOOP3_139093) and Sinergia Grant (CRSII5_180345), the Canton of Zürich, the UZH Foundation for Research in Science and the Humanities, the University of Colorado School of Medicine, and the Children's Hos-pital Colorado Foundation, NIH/NHLBI 1R01HL168097-01A1 (K.D.P, R.L., C.M.); C.M. holds the Helen and Arthur Johnson Chair for the Cardiac Research Director; the Institut Pasteur and the Agence Nationale de la Recherche Laboratoire d'Excellence Revive, Investissement d'Avenir (ANR-10-LABX-73) (S.T.).

## Author contributions

E.H. was involved in conceptualization, methodology, investigation, validation, visualisation, writing of the original draft, project adminis-tration and funding acquisition. H.D. and F.S.G. carried out CT scan analysis and visualisation, writing review and editing. K.D.P and R. L. performed experimental procedures, writing review and editing. F.L. completed microscopy acquisitions. C.M. and A.H. provided biological materials and completed writing review and editing. S.T. was involved in conceptualization, writing review, editing and funding acquisition.

## Competing interests

The authors declare no competing interests.
