## [Peer Review file · Nature Communications]

Co-option of neck muscles supported the vertebrate water-to-land transition

Corresponding Author: Dr Eglantine Heude

Version 0:

Reviewer comments:

Reviewer #1

(Remarks to the Author)

In this manuscript Heude et al. provide a very comprehensive yet concise analysis and synthesis of the evolutionary trajectory of muscles at the head-trunk interface associated with the evolution of a neck during the fish to tetrapod transition. Two sets of data are presented. The first set (Fig. 1 and 2) uses genetic lineage tracing in zebrafish to visualize the respective contributions of trunk mesoderm (*pax3* expressing) and cardiopharyngeal mesoderm (CPM) (*tbx1* expressing) to the head trunk interface muscles. This is further complemented by visualising the innervation pattern of cranial nerves. This strategy identifies the cucullaris muscle (the main muscle group to connect head and trunk) and shows a lineage tracing pattern consistent with CPM and not trunk muscle origin, while hypobranchial muscles are identified as being trunk derived. This confirms that these muscle groups have similar developmental origins in zebrafish as in tetrapods.

The second set of data (Fig 3. and supplement) is a comprehensive analysis of uCT data in zebrafish, bichir, lungfish, coelacanth salamanders and Anolis. This data set goes to show the presence or absence of hypobranchial, cucullaris, coracobranchial and laryngeal muscles in a cogently selected taxon sample relevant to the osteichthyes. This analysis confirms an ancestral origin of the various muscle groups and shows that absence of the laryngeal muscles in zebrafish is secondarily derived, presumably due to loss of air breathing, thereby strongly supporting an original function of the laryngeal muscles in a respiratory capacity. Combined with the lineage tracing data the authors propose a model whereby during the fish to tetrapod transition, the CPM derived cucullaris expanded from head to trunk to encompass and stabilise the neck, and conversely trunk musculature expanded into the head to form the tongue.

I find the manuscript extremely clean, interesting and well written. The final synthesis of the manuscript (Fig.4) certainly makes a text book worthy contribution.

I also have some comments.

- The Cre of particularly *tbx1* does not seem to be extremely efficient and as I can judge by the images in Fig. 2 there appears to be quite some mosaicism. In the B panels where the cucullaris is to be identified, it struck me that in the merged image where yellow is presumably to indicate presence of both MyHC and GFP (also it would be good to better indicate this in the legend so that no confusion can arise to this point) exactly the cc is not yellow but mosaic in purple and green, suggesting that not the same cells could be labeled – or perhaps it is just an artefact of the image processing settings. Nevertheless, since this image critically supports one of the main claims of the paper, I think it is necessary to clarify this. Also, as a reader it is often very hard to assess whether the images shown are more or less randomly chosen out many very similar experiments, or that they are the very best out of a few times that things finally worked. In this context it is relevant to report how often this was reproduced and perhaps introduce a figure with reproduced results in the supplement. (I apologize if I overlooked the numbers somewhere in an appendix, in that case it is better to make them prominently visible, for instance in the figure legend).

- In the paper, the authors build onto a large body of comparative research detailing and speculating about the homology of muscle groups in fish that goes back to the 19th century. Inevitably therefore, there is overlap with the findings of the authors and findings/speculations/hypotheses that have been published before, which are I think also mostly cited in the manuscript. Still, as some places I however have a bit of difficulties is to understand what exactly is completely novel; what is simply confirming previous observations (and why the observations of the authors are not just repeating previous findings but

actually provide a higher level of confidence in the final conclusions); if hypotheses are brought forward completely de novo or are built upon prior suggestions. An example I can give is for instance the paragraph of line 230-237, where the first sentence states "Interestingly, our data show that the bichir, coelacanth, and lungfish possess a laryngeal musculature." Raising the impression with the reader that these are all novel findings. Followed by "a previous ontogenetic morphological analysis...supports a CPM origin of the laryngeal musculature in the early diverging Australian lungfish *Neoceratodus*" indicating that in fact the presence of laryngeal muscles in lungfish was uncontroversial. The paper is laudably concise, but perhaps some more space can be dedicated to elaborate in this context.

- I find the yellow shade used to label the pectoral girdle in Fig. 2J/ Fig. 4 hard to distinguish. Please critically review/revise for better accessibility.

Joost Woltering

Reviewer #2

(Remarks to the Author)

The separation of the skull from the pectoral girdle, which led to the development of a functional neck, required significant changes in the musculoskeletal system. Despite the clear origins of head and trunk muscles from distinct embryonic mesoderm populations, the evolutionary origins of neck muscles at the head-trunk junction remain unclear. In this study, the authors explored the evolution of the vertebrate neck by integrating experimental embryology with comparative anatomy. Based on their observations, the authors proposed the model in which muscle groups linking the head and trunk share conserved mesodermal origins in both fishes and tetrapods. Furthermore, they proposed that the expansion of mesodermal cells into the head and trunk domains during embryonic development was crucial for the emergence and adaptation of the tetrapod neck.

In this paper, the authors address a longstanding and significant issue concerning the origins of neck muscles in vertebrates. The hypothesis they propose is indeed intriguing; however, their experimental data fall short of providing a robust foundation for their hypothesis for the following reasons.

The most pivotal findings presented in this paper is the demonstration that cucullaris muscles (cc) and coracobranchial muscles (cbm) in the zebrafish *tbx1*-lineage GFP reporter line are GFP positive and innervated by the vagus nerve X (nX). However, the data provided do not convincingly support these findings. The muscle designated as cc in Figure 2 B' is challenging to discern as GFP-positive in Figure 2 B", and the cbm is only partially GFP positive in Figure 2C". I suggest the authors to show clearer images showing GFP signals in cc and cbm, and utilize specimens where cc and bl are distinctly separated, as in Figure 1E. Enlarged pre-3D raw data images showing GFP signals in cc and cbm are also imperative. The discrepancy in the shape of cc between the 3D image in Figure 2E and Figure 2B raises questions; if different specimens were employed, ensuring consistency in the specimen used for 3D imaging is necessary. In addition, a finer depiction of nerves would enhance clarity in the neural innervations. If cc and cbm are indeed GFP-positive, 3D images of these GFP-positive regions should also be included. Pre-3D raw data on Ac Tu, MyHC and GFP as well as histological sections clearly depicting neural innervations to each muscle, are essential. If muscle definitions are based on neural innervations, proofs of such innervations via retrograde labeling or similar methods is necessary. Furthermore, the criteria for coloring nerves and muscles in 3D images should be explicitly stated.

Concerning the anatomical data presented in Figure 3, the absence of pre-3D raw data or clear criteria for defining each muscle raises questions about the rationale behind the authors' conclusions, reducing the figure to what may appear as arbitrarily colored illustrations. It is crucial to exhibit pre-3D raw data for the cucullaris musculature, coracobranchial musculature, laryngeal musculature, as well as nX across all specimens to substantiate the draw conclusions. The convincing images showing the exact innervation sites of nX are ambiguous; if nX innervation is used for defining each muscle, evidence (i.e. histological sections or retrograde analyses) of nX innervating the defined muscle across all samples is requisite. Furthermore, please also include the model name of the synchrotron CT scan used and the imaging conditions. Finally, given the extensive research on the muscle patterns of the vertebrate species under discussion, the authors are urged to elucidate why some of their findings diverge from previous studies.

Reviewer #4

(Remarks to the Author)

This is a great paper aiming to reveal the embryological origin of of head-neck transition muscles.

As cucullaris was in several studies already shown to be a branchial arch muscle it is (for me) not surprising that it is CPM derived, but it is great to see this in such clear data and figure plates.

Overall there are mostly tiny things to remark as indicated in the PDF.

Minor suggestions:

One thing I would like the authors to think about is the use of cucullaris throughout the investigated study. As soon as

branchial arch levators are identifiable the remaining shoulder-girdle to head muscle is the protractor pectoralis (Ziermann et al. 2014: Zoological Journal of the Linnean Society, 172(4), 771-802). This protractor pectoralis gives then rise to trapezius and sternocleidomastoid. That would mean for this study that all animals have cucullaris-derived muscles. That is ZF already has protractor pectoralis, as well as the other fishes and salamanders mentioned in this paper and the amniotes have sternocleidomastoid and trapezius. I'm not against using cucullaris, but it would be more precise to call them cucullaris-derived muscles. This is true for the figure legends, too.

All figures are amazing. In figure 1 cc (cucullaris muscle) is mentioned in the figure legend but not marked any of the figure parts. Unfortunately, I wasn't able to watch the video (didn't start), but I'm sure it is also great.

Version 1:

Reviewer comments:

Reviewer #1

(Remarks to the Author)

The authors have addressed all my previous concerns, in particular I find the new *tbx1*-Cre lineage tracing data very convincing.

Reviewer #2

(Remarks to the Author)

Thank you for addressing several of the comments and concerns raised during the initial review process. However, I find that some crucial aspects remain unsolved. I would like to highlight a few key points that I believe are crucial for the integrity of the conclusions.

1. Data for zebrafish: In my previous review, I emphasized the importance of clearly depicting neural innervations in the 3D images, particularly given that the definition of the cucullaris muscles (cc) hinges on their innervation by the vagus nerve (nX). Specifically, I stated, "Pre-3D raw data on Ac Tu, MyHC, and GFP as well as histological sections clearly depicting neural innervations to each muscle, are essential. If muscle definitions are based on neural innervations, proofs of such innervations via retrograde labeling or similar methods is necessary." Despite authors' response that current methods are optimal, the images provided, including Figure 2k and the corresponding movie (0:11~0:12), do not convincingly demonstrate the nX innervations in the cucullaris muscles. The cc appears to be merely adjacent to the nX, rather than being innervated by it. This ambiguity raises concerns about the accuracy of the muscle definitions. To address this, I strongly recommend including pre-3D raw data or histological sections that explicitly show the nX innervating the cc. I expect authors to provide images of similar quality to those in Figure 3F of (Higashijima et al., 2016, J. Morph.). Without such evidence, the conclusions drawn regarding muscle identity remain speculative.

2. Data for the Bichir, Lungfish, Coelacanth, Axolotl, Newt, and Lizard: The authors' response indicates that obtaining additional samples is not feasible due to limitations in the availability of bichir, coelacanth, lungfish. While I understand these constraints for coelacanth and lungfish, other species (bichir, axolotl, newt, and lizard) are readily available through various suppliers, including pet shops and aquariums. Therefore, it is entirely feasible to obtain these specimens for further analysis. The availability of these samples negates the justification for not performing additional experiments, particularly when such data could significantly strengthen the conclusions of the study. As I stated in my previous review, it is critical to provide more robust evidence for nX innervation if this is a central criterion for muscle definition. For instance, in the supplementary figures, the images provided do not clearly delineate the cc and bone boundaries (Supplementary Fig. 6b) or show definitive innervation by nX (Supplementary Fig. 3b, 4b, 6b, 7b, 8b). Additionally, synchrotron imaging and the reconstruction of these images are time-consuming and that the authors have analyzed only a single specimen for each sample. Given that images were obtained from specimens that were fixed post-mortem, it is possible that the cucullaris muscle may have been compromised, especially considering that these muscles consist of only a few myofibers. Although the trapezius muscle in lizard is anatomically clear and additional experiment may not be required, given the availability of other samples (bichir, axolotl, and newt), as well as antibodies for muscle and nerve markers, I believe that further efforts should be made to present supporting data that validate the muscle definitions. If histological sections are indeed inadequate, as authors have suggested, whole-body immunolabeling in fish and amphibian juveniles (Pende et al., 2020 Sci Adv 6, aba0365) should be considered as a feasible alternative. This would significantly strengthen the credibility of the study.

In conclusion, while this study addresses a significant and longstanding question in vertebrate evolutionary biology, the robustness of the findings hinges on the clear depiction and verification of neural innervations. I urge authors to consider the suggestions outlined above to enhance the rigor and reliability of the conclusions.

Reviewer #4

(Remarks to the Author)

The authors have addressed all my previous comments. They now highlight clearer novel insights based on their research.

They clarified the text by adding 'cucullaris-derived' where applicable for muscles. The new included figures (in main figures and supplementary) support the reproducibility of the research. This time, I was also able to watch both videos.

I have only some minor suggestions:

- Line 64: 'muscles that connect to the hyoid bone ventrally' – add 'to'
- Line 66: laterally
- Line 94: you wrote 'the vagal nerve (cranial nerve CN X)' and then in line 97 'accessory nerve (CN XI) derived from the vagus nerve 4,6,29'. I would suggest to add (CN X) after nerve in line 97 for readers that are not familiar with cranial nerve terminology
- Line 110, 124: 'connecting the head to the trunk' – exchange 'to' with 'and'
- Line 121: 'innervation, the attachments to skeletal' – add 'and' after comma
- Line 164: 'The vagus nerve also innervates the coracobranchial muscles ventrally" suggestion to change to 'The vagus nerve also innervates the ventrally located coracobranchial muscles' – Reason: the original sentence could be misunderstood in the way that the nerve approaches the muscle from ventral
- Lines 199-200: 'ceratohyal (hyoid arch) and the mandible, respectively' – suggestion to add '(mandibular arch)' after mandible
- Line 207: 'hypobranchial, laryngeal and cucullaris-derived' – add comma after 'laryngeal'
- Line 297: add comma after))
- Line 571: add comma after coracobranchial

RESPONSE TO REVIEWERS

Major changes include:

- concerns of Reviewers #1 and #2 with additional work to modify the content of Fig. 2a-i.
- a new Supp. Fig. 1 that complements data in Fig. 2, and new panels in Supp. Fig. 2e-l
- two Supp. Movies with associated legends:
 - * Supplementary Movie 1. 3D interactive view of the neuromuscular system at the head/trunk transition of a 5 dpf zebrafish larva to complete data shown in Fig. 2j-l and Supplementary Fig. 2a-d.
 - * Supplementary Movie 2. 3D interactive view of the neuromuscular system at the head/trunk transition of a 7 dpf zebrafish larva to complete data shown in Supplementary Fig. 2e-l.
- new raw data in Supp. Figs. 3-9b
- all figures have been formatted according to the journal's standard

Reviewer #1:

In this manuscript Heude et al. provide a very comprehensive yet concise analysis and synthesis of the evolutionary trajectory of muscles at the head-trunk interface associated with the evolution of a neck during the fish to tetrapod transition. Two sets of data are presented. The first set (Fig. 1 and 2) uses genetic lineage tracing in zebrafish to visualize the respective contributions of trunk mesoderm (pax3 expressing) and cardiopharyngeal mesoderm (CPM) (tbx1 expressing) to the head trunk interface muscles. This is further complemented by visualising the innervation pattern of cranial nerves. This strategy identifies the cucullaris muscle (the main muscle group to connect head and trunk) and shows a lineage tracing pattern consistent with CPM and not trunk muscle origin, while hypobranchial muscles are identified as being trunk derived. This confirms that these muscle groups have similar developmental origins in zebrafish as in tetrapods.

The second set of data (Fig 3. and supplement) is a comprehensive analysis of uCT data in zebrafish, bichir, lungfish, coelacanth salamanders and Anolis. This data set goes to show the presence or absence of hypobranchial, cucullaris, coracobranchial and laryngeal muscles in a cogently selected taxon sample relevant to the osteichthyes. This analysis confirms an ancestral origin of the various muscle groups and shows that absence of the laryngeal muscles in zebrafish is secondarily derived, presumably due to loss of air breathing, thereby strongly supporting an original function of the laryngeal muscles in a respiratory capacity. Combined with the lineage tracing data the authors propose a model whereby during the fish to tetrapod transition, the CPM derived cucullaris expanded from head to trunk to encompass and stabilise the neck, and conversely trunk musculature expanded into the head to form the tongue.

I find the manuscript extremely clean, interesting and well written. The final synthesis of the manuscript (Fig.4) certainly makes a text book worthy contribution.

I also have some comments.

- The Cre of particularly *tbx1* does not seem to be extremely efficient and as I can judge by the images in Fig. 2 there appears to be quite some mosaicism. In the B panels where the cucullaris is to be identified, it struck me that in the merged image where yellow is presumably to indicate presence of both MyHC and GFP (also it would be good to better indicate this in the legend so that no confusion can arise to this point) exactly the cc is not yellow but mosaic in purple and green, suggesting that not the same cells could be labeled – or perhaps it is just an artefact of the image processing settings. Nevertheless, since this image critically supports one of the main claims of the paper, I think it is necessary to clarify this.

Also, as a reader it is often very hard to assess whether the images shown are more or less randomly chosen out many very similar experiments, or that they are the very best out of a few times that things finally worked. In this context it is relevant to report how often this was reproduced and perhaps introduce a figure with reproduced results in the supplement. (I apologize if I overlooked the numbers somewhere in an appendix, in that case it is better to make them prominently visible, for instance in the figure legend).

We thank the Reviewer for their positive assessment of our work.

The Cre-mediated recombination of the *tbx1*-lineage GFP reporter with OHT treatment showed mosaicism in the first version of the study. Therefore, we have optimized the experimental approach with longer OHT treatment from 6 hpf to 30 hpf (instead of 6 hpf to 16 hpf) and with extended recovery after heat-shock GFP-switch induction (4-5 h instead of 2-3 h); we have modified the Methods section accordingly. This resulted in higher recombination efficiency and redundancy in branchial arch derivatives including the cucullaris. We finally obtained at least 10 specimens with clear GFP signal in the cucullaris myofibers at 5 dpf; the number of positive specimens analysed is now specified in Figure legends (Figs. 1-2). Moreover, we have added a Supplementary Fig. 1 showing control and several positive reporter specimens (#1 to #5) to demonstrate recombination specificity, efficiency, and reproducibility.

- In the paper, the authors build onto a large body of comparative research detailing and speculating about the homology of muscle groups in fish that goes back to the 19th century. Inevitably therefore, there is overlap with the findings of the authors and findings/speculations/hypotheses that have been published before, which are I think also mostly cited in the manuscript. Still, as some places I however have a bit of difficulties is to understand what exactly is completely novel; what is simply confirming previous observations (and why the observations of the authors are not just repeating previous findings but actually provide a higher level of confidence in the final conclusions); if hypotheses are brought forward completely de novo or are built upon prior suggestions. An example I can give is for instance the paragraph of line 230-237, where the first sentence states “Interestingly, our data show that the bichir, coelacanth, and lungfish possess a laryngeal musculature.” Raising the impression with the reader that these are all novel findings. Followed by “a previous ontogenetic morphological analysis...supports a CPM origin of the laryngeal musculature in the early diverging Australian lungfish *Neoceratodus*” indicating that in fact the presence of laryngeal muscles in lungfish was uncontroversial. The paper is laudably concise, but perhaps some more space can be dedicated to elaborate in this context.

We agree that some aspects were equivocal in the previous version of the manuscript. We have now modified and completed some paragraphs in the Discussion section to clarify the novelty of our work and the uncertainties in the literature.

- I find the yellow shade used to label the pectoral girdle in Fig. 2J/ Fig. 4 hard to distinguish. Please critically review/revise for better accessibility.

The color code of the pectoral girdle has been modified to dark grey instead of the yellow shade in Figures 2 and 4 for better visualisation.

Reviewer #2

The separation of the skull from the pectoral girdle, which led to the development of a functional neck, required significant changes in the musculoskeletal system. Despite the clear origins of head and trunk muscles from distinct embryonic mesoderm populations, the evolutionary origins of neck muscles at the head-trunk junction remain unclear. In this study, the authors explored the evolution of the vertebrate neck by integrating experimental embryology with comparative anatomy. Based on their observations, the authors proposed the model in which muscle groups linking the head and trunk share conserved mesodermal origins in both fishes and tetrapods. Furthermore, they proposed that the expansion of mesodermal cells into the head and trunk domains during embryonic development was crucial for the emergence and adaptation of the tetrapod neck.

In this paper, the authors address a longstanding and significant issue concerning the origins of neck muscles in vertebrates. The hypothesis they propose is indeed intriguing; however, their experimental data fall short of providing a robust foundation for their hypothesis for the following reasons.

The most pivotal findings presented in this paper is the demonstration that cucullaris muscles (cc) and coracobranchial muscles (cbm) in the zebrafish *tbx1*-lineage GFP reporter line are GFP positive and innervated by the vagus nerve X (nX). However, the data provided do not convincingly support these findings. The muscle designated as cc in Figure 2 B' is challenging to discern as GFP-positive in Figure 2 B'', and the cbm is only partially GFP positive in Figure 2C''. I suggest the authors to show clearer images showing GFP signals in cc and cbm, and utilize specimens where cc and bl are distinctly separated, as in Figure 1E. Enlarged pre-3D raw data images showing GFP signals in cc and cbm are also imperative. The discrepancy in the shape of cc between the 3D image in Figure 2E and Figure 2B raises questions; if different specimens were employed, ensuring consistency in the specimen used for 3D imaging is necessary. In addition, a finer depiction of nerves would enhance clarity in the neural innervations. If cc and cbm are indeed GFP-positive, 3D images of these GFP-positive regions should also be included. Pre-3D raw data on Ac Tu, MyHC and GFP as well as histological sections clearly depicting neural innervations to each muscle, are essential. If muscle definitions are based on neural innervations, proofs of such innervations via retrograde labeling or similar methods is necessary. Furthermore, the criteria for coloring nerves and muscles in 3D images should be explicitly stated.

As specified for Reviewer #1, we have completed additional work by optimising the experimental approach to clarify the *tbx1* lineage data. We now show reproducible GFP signal in the cucullaris and coracobranchial myofibers by confocal acquisitions and maximum intensity projections (Fig. 2a-l, Supplementary Fig. 1h-s) and 3D renderings (Supplementary Fig. 1a-c).

In the present study we decided to use lightsheet microscopy to study muscular innervation in the zebrafish larva. This approach permits access, at high 3D resolution (voxel 0,650µm), the relative conformation of the nervous and muscular microscopic components in the specimens analysed. The cucullaris and coracobranchial muscles of interest consist of only a few myofibers, thereby limiting proper histological analysis of neuromuscular connections on sections, and restricting retrograde labelings due to technical and physical constraints. Thus, we believe that our experimental approach is the most relevant for analysis of the specimens.

To complete the data collected in 5 dpf control larvae (Fig. 2j-l, Supplementary Fig. 2a-d, Supplementary Movie 1), we have performed equivalent stainings, acquisitions and renderings later during neuromuscular system differentiation at 7 dpf (Supplementary Fig. 2e-l, Supplementary Movie 2). Our data now show more clearly a branch of the vagus nerve connecting the cucullaris in zebrafish, adding experimental evidence and confirming its embryonic origin that we demonstrate by genetic expression and lineage analyses.

Concerning the anatomical data presented in Figure 3, the absence of pre-3D raw data or clear criteria for defining each muscle raises questions about the rationale behind the authors' conclusions, reducing the figure to what may appear as arbitrarily colored illustrations. It is crucial to exhibit pre-3D raw data for the cucullaris musculature, coracobranchial musculature, laryngeal musculature, as well as nX across all specimens to substantiate the draw conclusions. The convincing images showing the exact innervation sites of nX are ambiguous; if nX innervation is used for defining each muscle, evidence (i.e. histological sections or retrograde analyses) of nX innervating the defined muscle across all samples is requisite.

We have completed Supplementary Figs. 3-9 with CT scan raw data including pre- and post-otic virtual frontal sections showing the structures of interest that we segmented for 3D reconstructions. As stated at the beginning of the Results section, muscle identity is defined by the embryonic mesodermal origin, the innervation, and the attachments to skeletal structures. In the comparative anatomy analysis, the main factor and traditional way to define muscle identity involves muscle connections to skeletal components (see for reference Dearden *et al.* 2021 *J. Anat.*). We have segmented the nX in the bichir, coelacanth and lungfish as a second argument to confirm identity when embryological data or description in the literature are unclear or missing. The latter species analysed in the present study constitute rare (bichir, lungfish juveniles) to unique (coelacanth juvenile) specimens worldwide, and histological sections and retrograde analyses are therefore not experimentally feasible.

Furthermore, please also include the model name of the synchrotron CT scan used and the imaging conditions.

We have now added the model name in the Methods section and tables in Supplementary Information reporting the parameters used for micro- and synchrotron CT scan imaging (Supplementary Tables 1-2).

Finally, given the extensive research on the muscle patterns of the vertebrate species under discussion, the authors are urged to elucidate why some of their findings diverge from previous studies.

We thank the Reviewer for raising this important point. The embryonic origin of neck muscles has been a subject of controversy due to the location of its embryonic precursors at the interface of cardiopharyngeal, somitic, and lateral plate mesoderm populations, aspects that we discuss in the Introduction section of the manuscript. As described for Reviewer #1, we have now clarified our findings in regards to previous studies in the Discussion section.

Reviewer #4

This is a great paper aiming to reveal the embryological origin of head-neck transition muscles.

As cucullaris was in several studies already shown to be a branchial arch muscle it is (for me) not surprising that it is CPM derived, but it is great to see this in such clear data and figure plates. Overall there are mostly tiny things to remark as indicated in the PDF.

We thank the Reviewer for their positive assessment of our work.

We have considered the comments and suggestions indicated in the PDF and modified the text accordingly.

Minor suggestions:

One thing I would like the authors to think about is the use of cucullaris throughout the investigated study. As soon as branchial arch levators are identifiable the remaining shoulder-girdle to head muscle is the protractor pectoralis (Ziermann et al. 2014: Zoological Journal of the Linnean Society, 172(4), 771-802). This protractor pectoralis gives then rise to trapezius and sternocleidomastoid. That would mean for this study that all animals have cucullaris-derived muscles. That is ZF already has protractor pectoralis, as well as the other fishes and salamanders mentioned in this paper and the amniotes have sternocleidomastoid and trapezius. I'm not against using cucullaris, but it would be more precise to call them cucullaris-derived muscles. This is true for the figure legends, too.

We believe that there might be some confusion in the terminology. We used the cucullaris name to facilitate comparison between the vertebrate species analysed. We agree that the use of *cucullaris derivatives* or *cucullaris-derived muscles* is more accurate when comparing species from different clades; we have modified the text accordingly. However, we have kept the term cucullaris when referring to the embryonic anlagen or to the overall muscle group *cucullaris musculature* (as for

other muscle groups such as *laryngeal* or *hypobranchial musculature*) in the color code panels presented in Figs 3-4 and Supplementary Figs 3-9.

All figures are amazing. In figure 1 cc (cucullaris muscle) is mentioned in the figure legend but not marked any of the figure parts.

In Figure 1, the cucullaris is indicated in panel (m).

Unfortunately, I wasn't able to watch the video (didn't start), but I'm sure it is also great.

As indicated for Reviewers #2, we have now added a new movie (Supplementary movie 2) showing the connection of the vagal nerve to the cucullaris at 7 dpf to complete data previously presented only at 5 dpf (Supplementary movie 1). We hope that the current mp4 versions of the videos will work correctly and if not, we can provide another coded format.

RESPONSE TO REVIEWERS

Major changes include:

- concerns of Reviewer #2 with additional panels in Supp. Fig. 2m-o, a new Supp. Fig. 3 with associated video (Supp. Movie 3) and corresponding captions
- Some text modifications following comments from Reviewer #2 and Reviewer #4 and from the Editor

Reviewer #1:

The authors have addressed all my previous concerns, in particular I find the new *tbx1*-Cre lineage tracing data very convincing.

We thank the Reviewer for the positive assessment of our work.

Reviewer #2

Thank you for addressing several of the comments and concerns raised during the initial review process. However, I find that some crucial aspects remain unsolved. I would like to highlight a few key points that I believe are crucial for the integrity of the conclusions.

1. Data for zebrafish: In my previous review, I emphasized the importance of clearly depicting neural innervations in the 3D images, particularly given that the definition of the cucullaris muscles (cc) hinges on their innervation by the vagus nerve (nX). Specifically, I stated, "Pre-3D raw data on Ac Tu, MyHC, and GFP as well as histological sections clearly depicting neural innervations to each muscle, are essential. If muscle definitions are based on neural innervations, proofs of such innervations via retrograde labeling or similar methods is necessary." Despite authors' response that current methods are optimal, the images provided, including Figure 2k and the corresponding movie (0:11~0:12), do not convincingly demonstrate the nX innervations in the cucullaris muscles. The cc appears to be merely adjacent to the nX, rather than being innervated by it. This ambiguity raises concerns about the accuracy of the muscle definitions. To address this, I strongly recommend including pre-3D raw data or histological sections that explicitly show the nX innervating the cc. I expect authors to provide images of similar quality to those in Figure 3F of (Higashijima et al., 2016, *J. Morph.*). Without such evidence, the conclusions drawn regarding muscle identity remain speculative.

We thank the Reviewer for raising these important points.

To address the concerns and according to Editor's suggestions we have completed Supp. Fig. 2 with orthoslice views of raw images (panels m-o) that now clearly show on a single plane one of the branchial branches of the vagus nerve connecting to the cucullaris. We hope that these new perspectives combined with our data on the embryonic origin and skeletal connections will fully convince the community on the cucullaris identity and homology in zebrafish.

2. Data for the Bichir, Lungfish, Coelacanth, Axolotl, Newt, and Lizard: The authors' response indicates that obtaining additional samples is not feasible due to limitations in the availability of bichir, coelacanth, lungfish. While I understand these constraints for coelacanth and lungfish, other species (bichir, axolotl, newt, and lizard) are readily available through various suppliers, including pet shops and aquariums. Therefore, it is entirely feasible to obtain these specimens for further analysis. The availability of these samples negates the justification for not performing additional experiments, particularly when such data could significantly

strengthen the conclusions of the study. As I stated in my previous review, it is critical to provide more robust evidence for nX innervation if this is a central criterion for muscle definition. For instance, in the supplementary figures, the images provided do not clearly delineate the cc and bone boundaries (Supplementary Fig. 6b) or show definitive innervation by nX (Supplementary Fig. 3b, 4b, 6b, 7b, 8b). Additionally, synchrotron imaging and the reconstruction of these images are time-consuming and that the authors have analyzed only a single specimen for each sample. Given that images were obtained from specimens that were fixed post-mortem, it is possible that the cucullaris muscle may have been compromised, especially considering that these muscles consist of only a few myofibers. Although the trapezius muscle in lizard is anatomically clear and additional experiment may not be required, given the availability of other samples (bichir, axolotl, and newt), as well as antibodies for muscle and nerve markers, I believe that further efforts should be made to present supporting data that validate the muscle definitions. If histological sections are indeed inadequate, as authors have suggested, whole-body immunolabeling in fish and amphibian juveniles (Pende et al., 2020 Sci Adv 6, aba0365) should be considered as a feasible alternative. This would significantly strengthen the credibility of the study.

In conclusion, while this study addresses a significant and longstanding question in vertebrate evolutionary biology, the robustness of the findings hinges on the clear depiction and verification of neural innervations. I urge authors to consider the suggestions outlined above to enhance the rigor and reliability of the conclusions.

To address the concerns of Reviewer #2 about sample size and tissue integrity, and following the Editor's suggestions, we have clarified these points in the results and method sections for more transparency.

Regarding the comparative anatomy analysis, the main proxy for defining muscle identity is the muscle connections to skeletal components (see for reference Dearden *et al.* 2021 *J. Anat.*). We have segmented the nX in the Bichir, Coelacanth and Lungfish as a second argument to confirm identity when embryological data or description in the literature were unclear or missing.

In contrast, the innervation and identity of cucullaris-derived muscles in the tetrapod species analyzed is already well documented in the literature and is not under debate.

However, as a tetrapod reference to complete data shown during zebrafish development, we have performed whole-mount immunolabelling of the neuromuscular system of an axolotl larva presented in a new Supp. Fig. 3 and Supp. Movie 3 to highlight the connection of the vagus nerve to the cucullaris as observed in fish.

Reviewer #4

The authors have addressed all my previous comments. They now highlight clearer novel insights based on their research. They clarified the text by adding 'cucullaris-derived' where applicable for muscles. The new included figures (in main figures and supplementary) support the reproducibility of the research. This time, I was also able to watch both videos.

I have only some minor suggestions:

- Line 64: 'muscles that connect to the hyoid bone ventrally' – add 'to'

- Line 66: laterally
- Line 94: you wrote 'the vagal nerve (cranial nerve CN X)' and then in line 97 'accessory nerve (CN XI) derived from the vagus nerve 4,6,29'. I would suggest to add (CN X) after nerve in line 97 for readers that are not familiar with cranial nerve terminology
- Line 110, 124: 'connecting the head to the trunk' – exchange 'to' with 'and'
- Line 121: 'innervation, the attachments to skeletal' – add 'and' after comma
- Line 164: 'The vagus nerve also innervates the coracobrachial muscles ventrally" suggestion to change to 'The vagus nerve also innervates the ventrally located coracobrachial muscles' – Reason: the original sentence could be misunderstood in the way that the nerve approaches the muscle from ventral
- Lines 199-200: 'ceratohyal (hyoid arch) and the mandible, respectively' – suggestion to add '(mandibular arch)' after mandible
- Line 207: 'hypobranchial, laryngeal and cucullaris-derived' – add comma after 'laryngeal'
- Line 297: add comma after))
- Line 571: add comma after coracobrachial

We are grateful to the Reviewer for the positive assessment of our work and careful reading of our manuscript.

We have considered the comments and suggestions and modified the text accordingly.

We hope that this revised version fulfills all comments we received from the Reviewers and Editor and is now suitable for publication.